# High-Speed Extraction of Regions of Interest in Optical Camera Communication Enabled by Grid Virtual Division

**DOI:** 10.3390/s22218375

**Published:** 2022-11-01

**Authors:** Xin Hu, Pinpin Zhang, Yimao Sun, Xiong Deng, Yanbing Yang, Liangyin Chen

**Affiliations:** 1Southampton Business School, University of Southampton, Southampton SO17 1BJ, UK; 2College of Computer Science, Sichuan University, Chengdu 610065, China; 3Institute for Industrial Internet Research, Sichuan University, Chengdu 610065, China; 4School of Information Science and Technology, Southwest Jiaotong University, Chengdu 611756, China

**Keywords:** optical camera communication (OCC), visible light communication (VLC), region of interest (RoI), multi-pixel search (MPS), grid virtual division (GVD)

## Abstract

Optical camera communication (OCC), enabled by light-emitting diodes (LEDs) and embedded cameras on smartphones, has drawn considerable attention thanks to the pervasive adoption of LED lighting and mobile devices. However, most existing studies do not consider the performance bottleneck of Region of Interest (RoI) extraction during decoding, making it challenging to improve communication capacity further. To this end, we propose a fast grid virtual division scheme based on pixel grayscale values, which extracts RoI quickly without sacrificing computational complexity, thereby reducing the decoding delay and improving the communication capacity of OCC. Essentially, the proposed scheme uses a grid division strategy to divide the received image into blocks and randomly sample several pixels within different blocks to quickly locate the RoI with high grayscale values in the original image. By implementing the lightweight RoI extraction algorithm, we experimentally verify its effectiveness in reducing decoding latency, demonstrating its superior performance in terms of communication capacity. The experimental results clearly show that the decoding delay of the proposed scheme is 70% lower than that provided by the Gaussian blur scheme for the iPhone receiver at a transmission frequency of 5 kHz.

## 1. Introduction

As the demand for ubiquitous connectivity and high capacity increases [1], traditional radio frequency (RF) wireless communications technologies are plagued by spectrum deficiency and high energy-consumption issues. As an alternative, visible light communication (VLC) opens up a new frontier for sixth-generation (6G) communication thanks to its superior large, cost-free optical spectrum and energy efficiency [2,3]. Along with this, the widespread adoption of LED lighting infrastructure and camera-equipped smart devices further motivates the usage of VLC. Particularly, optical camera communication (OCC), as a practical VLC technology that can reuse high-efficiency LEDs and camera-equipped smartphones for data communication, has attracted extensive attention from academic and industrial communities [4,5,6,7]. Compared with traditional RF wireless communication technologies, OCC has the advantages of non-licensed channels, no electromagnetic interference radiation and low power consumption [8,9,10], which can be used in the fields of indoor wireless communication [11,12], indoor precise positioning [13,14], underwater environments [15] and intelligent transportation [16,17].

Optical camera communication systems built on commercial off-the-shelf (COTS) equipment typically employ LED lighting infrastructure as transmitters and low-cost cameras as receivers. However, limited by the inherent conditions of low frame rate (typically limited to 30/60 frames per second (fps)) and the low sampling rate of ordinary cameras, the early works of OCC can only reach a data rate of several bytes per second [18,19]. To overcome this limitation, Prof. Haas’s team [4] proposed using the rolling shutter effect of complementary metal oxide semiconductor (CMOS) cameras to sample ON–OFF Keying (OOK) modulation signals, which can significantly increase the communication data rate to the kbps level [20,21,22]. During rolling shutter operation, instead of exposing the entire image frame at once, the camera conducts exposures in a column-by-column (or row-by-row) manner, forming bright or dark stripes in the received image illustrating the “ON” or “OFF” status of the LED transmitters [20,23]. In general, the camera first searches for the region of bright and dark stripes concentration from the received image; that is, the Region of Interest (RoI), and then extracts a single row (column) pixel from the RoI to form a grayscale vector for demodulation and decoding of optical signals. However, the existing OCC systems still have problems such as small communication capacity, which need to be resolved.

Aiming at addressing the problem of the small communication capacity of OCC, several methods have been proposed in prior works. T.H. Do and H. Lee et al. [24,25] systematically analyzed and revealed the optical communication reception capability of ordinary cameras from the theoretical and experimental levels, and focused on analyzing system communication capabilities using Frequency Shift Keying (FSK) modulation. P. Luo et al. [26] of Huawei proposed a series of undersampling modulation schemes, such as Undersampling Phase Shift ON–OFF Keying Modulation (UPSOOKM [27]), Undersampling Pulse Amplitude Modulation (UPAM [28]), to realize a flicker-free OCC system with higher spectral efficiency. However, due to the low frame rate of ordinary cameras and the low response frequency of ordinary LED luminaires, the rate of existing OCC systems based on low-order modulation is still meager. To this end, the researchers tried to improve the communication capacity of the OCC system by designing new modulation methods that support higher orders or using the multi-lamp collaborative transmission.

Since the camera can usually distinguish the color of the light, P. Hu et al. [29] used RGB LED lamps to realize high-order Color Shift Keying (CSK) modulation, which improved the communication capacity of OCC. In order to support the compatibility of communication and lighting, Y. Yang et al. [6] proposed a high-order modulation technology based on a combination switch. The technology uses the characteristics of ordinary LED lamps with multiple lamp beads to control the light beads on the LED lamp by grouping, forming a high-order modulation signal, thereby improving the communication ability. In addition, using the multi-lamp collaborative transmission is also an effective way to improve the performance of OCC systems. Y. Yang et al. [20,21] improved the communication capacity and reliability of the camera simultaneously through the coordinated transmission of multiple lamps by considering the characteristics of indoor multi-lamp lighting. The above work has improved the communication capacity and reliability of OCC to a certain extent. However, the above research has not considered the performance bottleneck of RoI extraction during the decoding process, which makes it difficult to further improve the communication capacity.

The traditional RoI extraction method first performs a grayscale conversion operation on the entire image received by the camera, followed by Gaussian blur and threshold discrimination to obtain the outline of the RoI. The decoder then extracts a row or column of pixels from within the contour to form a grayscale vector for subsequent demodulation and decoding to enable data communication. However, the realization of the Gaussian blur step needs to be filtered by building the corresponding weight matrix, which makes the entire Gaussian blur process very time consuming [30]. In addition, along with the rise of deep learning, some works are being carried out using the YOLO [31] algorithm to train models to extract the parts of interest through supervised training for RoI extraction in object recognition. There are also transfg [32] models based on the transformer that integrate all of the transformer’s raw attention weights into the attention graph to guide the network to efficiently and accurately select differentiated blocks of images and compute the relationships between them. However, a common problem with all these deep learning algorithms is that they take a long time to load the model and recognize the object, but RoI extraction in OCC does not require so much fine detection to figure out the outline of an LED transmitter. To cope with these issues, we propose a lightweight grid virtual division scheme based on pixel grayscale values, which extracts RoI quickly while maintaining low computational complexity. In particular, the proposed novel scheme randomly samples a number of pixels within the received image instead of the traversal of each pixel, thereby greatly improving the extraction efficiency. Extensive experiments on iOS smartphones strongly confirm the effectiveness of the proposed scheme. The decoding delay by the iPhone 8 Plus is 70% lower than those provided by the baseline under a transmission frequency of 5 kHz.

## 2. Background and Motivation

We set up the background of RoI extraction in this section. First, we briefly introduce the rolling shutter effect of the CMOS camera embedded in the intelligent device. Then, provide a detailed explanation of why the Gaussian blur steps used in the traditional RoI extraction process are time consuming.

### 2.1. Rolling Shutter

In rolling shutter mode, the photodiode does not collect all the light simultaneously, but instead scans the pixels sequentially using the CMOS image sensor (IS). Images are generated by row by row (or column by column) scanning and exposing one row (or column) of pixels at a time, with each row (or column) pixel in the sensor array having a different time to start and end the exposure. Thus, the delay in the exposure time allows us to record the relative structure of the target based on time. In this mode, different pixels obtained from the rolling shutter image sensor have different light signal intensities, which means that there are multiple bits of information. Precisely, the receiver can capture multiple switching states of the illumination source in the image. When the LED luminaire is “ON”, a bright stripe is generated in the image, representing the binary data “1”, and when the LED luminaire is “OFF”, a dark stripe is generated in the image, representing the binary data “0”, as shown in Figure 1. The faster the LED blinks, the greater the number of stripes and the narrower the width, so multiple bits of information can be encoded with this pattern [24,33].

### 2.2. Gaussian Blur

As mentioned previously, the rolling shutter mechanism adopts a sequential exposure method, displaying the “ON” and “OFF” status of the illumination source as stripes in the image. Only these stripes carry valid data information sent by the transmitter. Therefore, the extraction of the striped area (RoI) is a core factor affecting the performance of OCC. While one can quickly identify RoIs in images with the naked eye, automatically extracting them via a smartphone is by no means trivial due to the spatial distribution of transmitters. Existing work [30,34] typically uses computer vision (CV) technology to extract RoI, where Open Source Computer Vision (OpenCV) algorithms are integrated into demodulator Apps for smartphones.

As illustrated in Figure 2, the image frame received by the CMOS camera is first grayscaled. Subsequently, Gaussian blur and image binarization steps are used to obtain the RoI’s outline further. After the RoI outline in the receiving image is determined, the receiver decoder samples a row or column of pixel grayscale values from the RoI for subsequent demodulation and decoding to enable data communication. However, the realization of Gaussian blur requires filtering by constructing the corresponding weight matrix. The normal distribution is a significant weight distribution pattern, with larger values closer to the center of the bell curve and smaller values moving closer to the center of the bell curve. When calculating the average, we only need to take the “center point” as the origin, assigning different weights to each location according to the distance from the center pixel. Since the images are all two-dimensional, the density function of their two-dimensional normal distribution (also known as the Gaussian function) can be expressed as
(1)$$G\left(x,y\right)=\frac{1}{2\pi \sigma^{2}}e^{-\left(x^{2}+y^{2}\right)/2\sigma^{2}},$$
where *x* and *y* represent the horizontal ordinate distance from the point to be calculated to the center of the convolution kernel, respectively. σ represents the Gaussian function standard deviation, the influence of other pixels in the neighborhood on the center point of the convolution kernel.

According to the Gaussian function, the weight value of each pixel in the image received by the surveillance camera is calculated, and the weight matrix is normalized. For each pixel of the original communication image, the product of its neighborhood pixels and the corresponding elements of the weight matrix are calculated and then summed to obtain the Gaussian blur value of the current center pixel. The weight matrix is then shifted to the right to calculate the Gaussian blur value for each point. Obviously, the contour extraction technique in OpenCV needs to traverse each pixel in the received image, resulting in high computational complexity and time consumption during OCC decoding. However, most indoor lighting currently employs round, square or rectangular LED luminaires. It is relatively simple to extract the contours of these LED luminaires compared with the complex contour extraction in the field of CV. Therefore, we only need to iterate through part of the pixels rather than all of them for quick positioning, which will effectively diminish the decoding delay of the OCC receiver and strengthen the communication capacity of the OCC.

## 3. High-Speed RoI Extraction through Grid Virtual Division

In this section, we explain the overall design of the grid virtual division scheme based on the OCC system. We start with a brief overview of the OCC system workflow, and then we discuss its individual components in detail. Finally, we highlight the detailed steps of the grid virtual division scheme in the OCC receiver.

### 3.1. System Overview

As illustrated in Figure 3, the block diagram delineates the overall OCC architecture, which is partitioned into two main parts: a transmitter and a receiver. The transmitter commonly adapts high-efficiency white light LED luminaires to emit modulated light signals, and the circuit structure is simple and only uses ordinary transistors as drives. The receiver uses a variety of mobile devices with embedded CMOS cameras (such as tablets, laptops, and mobile phones) as its front end to receive the light signals emitted by the white LED. To cope with the issues in Section 1 and Section 2, we equip the receiver with the proposed scheme, which quickly identifies RoIs without sacrificing computational complexity and bit error rate (BER). The following subsections will be elaborated on the OCC transmitter, OCC receiver and the specific implementation of the proposed scheme.

### 3.2. OCC Transmitter

The light sources used in daily life include halogen lamps, incandescent light bulbs, fluorescent lamps and LEDs. Only LEDs are perfect light sources for optical camera communication due to their fast rate of ON–OFF switching. Therefore, we generally use white light LED luminaires as the transmitter device of OCC. The workflow diagram of the OCC transmitter is shown in the upper part of Figure 3. It is mainly composed of five parts: input information, encoding, a modulation circuit, an LED driver and LED luminaires. More specifically, the input information is first divided into blocks, and the block sequence number (BSN) is added before each block. Then, the corresponding encoding techniques are applied to maintain DC balance so as to avoid visible flicker. Finally, the encoded data are mapped to modulation symbols using a modulation technique, and the drive circuit controls the LED luminaires to emit the modulated light.

### 3.3. OCC Receiver

As illustrated in the lower part of Figure 3, the OCC receiver consists of six parts: CMOS camera, frame sampling, RoI extraction, pixel sampling, demodulator/decoding and output information. More specifically, the CMOS camera-equipped smartphone first uses the rolling shutter effect to capture images of the modulated light sources, forming bright and dark stripes in the received frame. The received frames then go through the Gaussian blur and image binarization in computer vision techniques to identify the contours of the ROI. Subsequently, the grayscale values of an intermediate row or column are sampled from the recognized RoIs to form a grayscale sequence. This grayscale sequence can be expressed as the following vector:(2)$$\vec{g}=\left[\begin{array}{ccccc} g_{i1} & g_{i2} & ... & g_{i(n-1)} & g_{in} \end{array}\right],$$
where $g_{ij}$ denotes the grayscale value of the *i*th row and *j*th column pixel. Finally, the grayscale information obtained after sampling is converted into logical data by setting thresholds to distinguish different grayscale value levels of the transmission signal.

### 3.4. Grid Virtual Division Scheme

RoI extraction is the core factor affecting communication performance, but the existing RoI extraction technique is computationally complex and time consuming. Taking these issues into account, we suggested a grid virtual division scheme that improves the efficiency of RoI extraction. Since the light emitted by the transmitter is brighter, the area where the RoI projected by the transmitter in the received image has a higher grayscale value of the pixels than the area where the non-RoI is located. Suppose a pixel with a high grayscale value is retrieved in a certain area, and there are still pixels with a high grayscale value in the positive and negative ten pixels on both sides of the point. In that case, it proves that the area is an RoI area with adjacent symbols. Therefore, the proposed scheme considers using the grayscale values of the pixels in the received image to quickly locate these RoIs. In this section, we employ three shapes of LED luminaire—round, square, and rectangular—as transmitters. The RoI projected by the transmitter carries desired information in a frame, as shown in Figure 4. In more detail, the proposed scheme can be divided into the following four steps, and the flow chart is shown in Figure 5.

Step 1—Image Chunking: When the CMOS camera receives a whole image containing optical communication information, the proposed scheme first establishes a virtual grid of the original image, as shown in the virtual grid composed of orange dotted lines in Figure 4. Generally, the partition block is divided into n×m blocks according to the resolution of the received image. In this section, we establish a grid with the partition size of 6×6 as an example, as shown in Figure 4. It is worth noting that this step only divides the received image virtually into 6×6 parts, without traversing the entire image to divide it into 6×6 subgraphs. This is to make the sampling area smaller while maintaining computational complexity so that blocks with higher grayscale values are found.

Step 2—Pixels Sampling: After the virtual grid is established, the decoder randomly samples the grayscale values of several pixels within the block from the uniform distribution. Since the area after grid division is small, if the RoI is located in that area, it is likely to produce a larger area of coverage within the block. Therefore, we sample the pixels in the left, middle and right areas of the middle row in each block to increase the hit rate of RoI detection in the block. To be specific, we first choose to randomly sample a pixel in the left area. If the pixel meets the requirements, we will directly save the block number and continue to process the next block. If the pixel does not meet the criteria, we will continue to perform the same operation in the following area until all three areas are completed. As mentioned before, the grayscale value in the area where the optical signal is concentrated is higher than that in the surrounding area. To better distinguish which sampled pixels are eligible, we set a grayscale threshold empirically so as to maximize recognition accuracy. For comparison with this threshold, in this step, we first store the sampled grayscale values in the collection, which are represented as follows:(3)$$R=\{g_{m,n}^{i}|0<i\leq k,1\leq m\leq H,1\leq n\leq W\},$$
where *R* represents a collection of grayscale values for randomly sampled pixels within different grids, $g_{m,n}^{i}$ represents the grayscale value of the pixels in column *n* of row *m* in the *i*th block, *k* represents the number of blocks divided into the original image, H×W represents the resolution size of the original image. Afterwards, we compare the grayscale values sampled in the collection to the threshold in turn, recording the number of blocks of pixels with higher grayscale values.

Benefiting from the advantages of the previous step, when sampling pixels in a certain block, it is only necessary to divide the rows and columns of the original image by the partition sizes *n* and *m* to locate this area quickly. To be precise, the pixels sampling step calls the SEARCHPIXEL and SAMPLE functions in Algorithm 1 after the virtual grid has been established in the frame. SEARCHPIXEL function filters the randomly sampled pixels in the SAMPLE function and stores the block number that meets the requirements in the results container. As mentioned before, we sample the pixels in the left, center and right areas of each block’s middle row. As illustrated in Figure 4, the cyan-marked numbers in the frame indicate the block number that satisfies the condition after calling the SEARCHPIXEL function and the SAMPLE function in Algorithm 1.
**Algorithm 1** Grid virtual division algorithm**Input:** Image, *n*, *m*, threshold**Output:** $Vector_{rect}$1:**function**SearchPixel(Image,n,m)2:      **for** $block_{num}=1\to n*m$ **do**3:            $pixel_{left}\leftarrow Sample\left(Image,block_{num},left\right)$4:            $pixel_{center}\leftarrow Sample\left(Image,block_{num},center\right)$5:            $pixel_{right}\leftarrow Sample\left(Image,block_{num},right\right)$6:            **if** $pixel_{left}$  or  $pixel_{center}$  or  $pixel_{right}>threshold$ **then**7:                 Store $block_{num}$ in container $result_{vec}$8:            **end if**9:            $block_{num}++$10:      **end for**11:      **return** $result_{vec}$12:**end function**13: 14:**function**ImageCompose($result_{vec},Image,n,m$)15:      $n\leftarrow result_{vec}.size$16:      **for** i=0→n **do**17:            **if** At least one of the upper, lower, left and right blocks of the $result_{vec}\left[i\right]$ block is stored in container $result_{vec}$ **then**18:                 Concatenate adjacent blocks19:                 Calculate the $rect_{x}$, $rect_{y}$, $rect_{width}$, $rect_{height}$ of the new combined block20:                 Store coordinates in container $Vector_{rect}$21:                 i++22:            **else**23:                 Calculate the $rect_{x}$, $rect_{y}$, $rect_{width}$, $rect_{height}$ of the original block24:                 Store coordinates in container $Vector_{rect}$25:                 i++26:            **end if**27:      **end for**28:      **return** $Vector_{rect}$29:**end function**30: 31:**function**Sample($Image,block_{num},location$)32:      h←Image.rows/n33:      w←Image.cols/m34:      **if** location==left **then**35:              $result\leftarrow random.sample(range\left(\left(block_{num}\%m-1\right)*w,\left(block_{num}\%m-1\right)*w+w/3\right),1)$36:      **else if** location==center **then**37:              $result\leftarrow random.sample(range\left(\left(block_{num}\%m-1\right)*w+w/3,\left(block_{num}\%m-1\right)*w+2w/3\right),1)$38:      **else**39:              $result\leftarrow random.sample(range\left(\left(block_{num}\%m-1\right)*w+2w/3,\left(block_{num}\%m-1\right)*w+w\right),1)$40:      **end if**41:      **return** result42:**end function**

Step 3—Multi-block Stitching: After obtaining the block number at the area where the optical signal is easily concentrated, this step first needs to determine the position relationship between the block number reserved in the previous step. If a complete ROI is segmented, the divided sub-RoIs must be adjacent. Therefore, it is possible to determine whether a block belongs to the following four situations according to this criterion:There are only upper and lower adjacent blocks;There are only left and right adjacent blocks;There are both upper and lower adjacent blocks and left and right adjacent blocks;There are neither upper and lower adjacent blocks nor left and right adjacent blocks.

More specifically, if the result of the current block number, plus or minus the number of columns, exists in the container, it is treated as the first case, because the upper and lower adjacent blocks are in the same column as the current block. For example, the result of the block number labeled “11” (current block number) plus “6” (number of columns) in Figure 4d exists in the container, so the block number “17” is the lower adjacent block of “11”. Therefore, this step then entails calling the IMAGECOMPOSE function in Algorithm 1 to store the coordinate orientation of the block merged by “11” and “17”. If there are subsequent consecutive block numbers for the current block number in the container, it is considered the second case. For example, the block marked “26” in Figure 4c has contiguous right adjacent blocks of “27”, “28”, and “29”, so it is necessary to call the IMAGECOMPOSE function in Algorithm 1 to store the coordinates of its constituent blocks. If all the above operation results are present, it is considered the third case. For example, the block marked “4” in Figure 4b has both a right adjacent block “5” and a lower adjacent block “10”. At the same time, its right adjacent block exists above block “11”, and its lower adjacent block exists on the left side of block “11”. Thus, this step needs to use the IMAGECOMPOSE function in Algorithm 1 to record the coordinate orientation of the block synthesized by these four blocks after determining the block situation. Finally, if none of the above results is present, it is considered the fourth case, in which no stitching operation is required. It is worth mentioning that parts of the RoI may be missed during the image combination process. For example, block number 22 in Figure 4b is not detected because the area occupied by the stripes is too small. To deal with this situation, after obtaining the coordinate orientation of the merged block, we need to extend a few pixels on both sides of the block before performing subsequent operations.

Step 4—Demodulation and Decoding: Once the coordinates are obtained, this step involves converting the grayscale to bits and recovering the transmitted data. To be more specific, the demodulator first samples a single row or column of pixels from the acquired area to form a grayscale sequence for demodulation. In order to identify the start of a transmission session, the demodulator first uses a coarse-grained threshold to distinguish the preamble, which is defined as several consecutive bright symbols in the packet. Subsequently, fine-grained thresholds are fitted using the grayscale information of the identified preambles to further convert the grayscale information between two adjacent preambles into logical data. Finally, the receiver recovers the sent messages and completes the decoding process of a whole frame.

In a nutshell, the above four-step implementation enables the OCC system to locate RoIs with high grayscale values without traversing all pixels. Since it is necessary to traverse the container storing the block numbers of high grayscale values, the time complexity of the algorithm can be calculated as O(n) according to the following equation:(4)T[n]=O(f(n)),
where T[n] denotes the execution time of the code, *n* denotes the size of the data and f(n) denotes the sum of the execution times of each line of code. Furthermore, the algorithm’s memory space required to be opened up does not vary with *n*, so the space complexity is O(1). From a subjective point of view, this scheme can effectively alleviate the decoding delay of the OCC receiver and promote the communication capacity of the OCC. This satisfies the functional assumption of Section 2. In order to draw the convinced conclusions, extensive experiments are also conducted in the subsequent sections to verify the robustness of the proposed scheme.

## 4. Results and Analysis

In this section, we first introduce hardware and software implements of the proposed OCC system. We then report the experimental results of the proposed OCC system with the grid virtual division scheme. To verify the effectiveness of the proposed scheme, we conclude by seriously discussing various metrics and providing some insights into further improvements of the performance of OCC.

### 4.1. Experiment Setup

In order to evaluate the performance of the proposed OCC system with the grid virtual division scheme, we build four prototype systems and demonstrate its experimental setup, as shown in Figure 6. At the transmitter side, we use two types of LED luminaires with different sizes and shapes (round LEDs with diameters of 0.13 m and 0.3 m, square LEDs with a side length of 0.14 m × 0.14 m and 0.25 m × 0.25 m) as transmitters in our proposed system. Specifically, we build the LED driver with commercial off-the-shelf components, which converts alternating current (AC) into direct current (DC) for further modulation via a low-cost MOSFET of SI2310A to drive the selected LED luminaires. An ARM Cortex-M4 GD32F330G8U6 microcontroller controls the MOSFET to generate modulation symbols, as shown in Figure 6a. The modulation method utilizes the ON–OFF Keying (OOK), thanks to its simplicity, without losing its generality. In addition, in order to make the data transmission more reliable, the original data are split into “packets” for batch transmission. At the receiver side, we take the CMOS camera of the iPhone 8 plus with 12-megapixels as the receiver and configure the smartphone camera to work in the preview mode with a frame rate of 30 fps. The exposure time of the camera is set to 1/8000 s, and the sensitivity (ISO) of the camera is set to 350. The detailed parameters are shown in Table 1. Last but not least, we demodulate and decode the optical communication signal through the built-in iOS application of the iPhone 8 Plus (debugged with Xcode 12.4) to complete the data communication between the OCC transmitter and the OCC receiver.

### 4.2. Grid Virtual Division Scheme Performance

Based on the prototype of the experimental system already deployed in Figure 6, this section first designs the comparison experiment between the traditional RoI extraction (as the benchmark) and the proposed schemes. In practical application scenarios, the effect of transmission frequency, LED shape or quantity on ROI extraction needs to be considered, so we set these three metrics to quantify the experimental results. We finally report the experimental results provided by the baseline and grid virtual division schemes under variable transmission frequency, variable LED shape and size and variable LED number, respectively. It is worth noting that more than 10,000 bits are recorded during each test to count the bit error rate (BER) of the proposed scheme at different data rates. Furthermore, we have calculated the processing time of machine-learning-based models (such as YOLO and Transformer). As the seventh-generation models in the YOLO series are more flexible and have faster processing times, we used the latest version of YOLO7 for testing and removed the loading times of the model to compare with the proposed method. However, YOLO7 still has a processing time of 12 ms after removing the loading time of the model, which is much longer than the selected baseline of Gaussian Blur, so we omit to report its results for following tests.

#### 4.2.1. Impact of Varying Transmission Frequencies

Since extraction time is a core indicator used to measure the efficiency of RoI extraction, in this subsection, we first explore the performance of the proposed scheme in terms of processing time under varying transmission frequencies. Specifically, the processing time is the elapsed time from when the receiver captures an image to when the RoI extraction is completed. In addition, we also examine the performance of the traditional scheme and the proposed scheme in terms of BER under varying transmission frequencies. We present the experimental results of each group and analyze them in the following sections.

We fix the receiver at a distance of 80 cm away from the No.1 transmitter (round LED with a diameter of 0.13 m) and compare the processing time of the proposed method with the baseline method under three transmission frequencies. We then represent the results obtained by the iPhone 8 Plus in Figure 7. Intuitively, the different frequencies have little effect on the extraction time of the RoI. However, our grid virtual division scheme is superior to the baseline at the same transmission frequency thanks to its advantage of not having to traverse all pixels. It is evident from the figure that the grid virtual division scheme substantially outperforms the baseline with its median times at 0.33 ms, 0.395 ms, and 0.335 ms under transmission frequencies of 4 kbps, 5 kbps, and 6 kbps, as the corresponding median times of the baseline are one order higher at 1.71 ms, 1.28 ms and 2.35 ms, respectively. This comparison evidently confirms the effectiveness of the proposed innovative scheme, which can extract RoI quickly while maintaining low computational complexity.

In addition, we also report the BER results obtained by iPhone 8 Plus under three different transmission frequencies, as shown in Figure 8. It can be seen that the BER of baseline and the proposed scheme experience a substantial increase to 0.72% and 0.81% when the transmission frequency reaches 6 kbps. This phenomenon can be explained by the fact that the strip width in the frame is shorter and hence lower symbol distance at higher transmission frequencies. Nonetheless, the grid virtual division scheme can achieve a reasonable BER below the FEC threshold of 0.38% at a maximum of 5 kbps while extracting RoIs quickly, which strongly confirms the feasibility and effectiveness of the proposed scheme. Additionally, a transmission frequency of 5 kbps is usually sufficient to support most OCC applications.

#### 4.2.2. Impact of Different LED Shapes and Sizes

As a practical communication, the various shapes and sizes of LEDs supported by OCC can have an impact on the efficiency of RoI extraction. Therefore, we evaluate the performance of the traditional and the proposed schemes in terms of processing time and BER based on LEDs of different shapes and sizes in this subsection. More specifically, we configure the transmission frequency at 5 kbps and fix the receiver at a distance of 80 cm away from the transmitter to measure the extraction efficiency of both schemes. As shown in Figure 6, the transmitters for the four prototypes consist of round LEDs with diameters of 0.13 m (R1) and 0.3 m (R2) and square LEDs with side lengths of 0.14 m (S1) and 0.25 m (S2), respectively. We present the experimental results obtained by the receiver using box plots as illustrated in Figure 9. It shows that compared with the baseline, the proposed scheme has better performance for different LED shapes and sizes. For round LED transmitters, the median times of 0.23 ms and 0.28 ms obtained by the grid virtual division scheme are significantly better than the corresponding median times of 1.28 ms and 1.26 ms for the baseline. For square LED transmitters, the median times of 0.44 ms and 0.41 ms calculated by the proposed scheme are significantly better than those of 1.99 ms and 0.99 ms for the baseline. It can be seen that the grid virtual division scheme has significant potential to improve the efficiency of RoI extraction.

Similar to the work in Section 4.2.1, we also count the BER results obtained by the receiver when replacing the transmitter with different shapes and sizes, as shown in Figure 10. It is evident from the figure that the BER calculated by the proposed scheme is slightly higher than that calculated by the baseline in the case of R1, R2 and S2. Specifically, the BER variation of the grid virtual division scheme ranges from 0.2% to 0.38% in the case of R1, R2 and S2, while the variation in the baseline BER in these four cases ranges from 0.12% to 0.34%. Nevertheless, the proposed scheme improves the efficiency of RoI extraction while maintaining a reasonable BER below the FEC threshold of 0.38%. Furthermore, we notice that the BER of S1 is slightly higher than the other three cases. This may be caused by the small size of the S1 and the fact that the cloth in front of the lamp absorbs some of the light intensity. Meanwhile, S2 has a more significant number of light strips in the lamp, so the BER is lower than that of S1.

#### 4.2.3. Impact of Different Numbers of LED

As mentioned earlier, the number of transmitters projected in a frame affects the extraction efficiency of the RoI to a certain extent. Therefore, in this subsection, we explore the performance of the traditional and proposed schemes under variable LED numbers. We still configure the transmission frequency at 5 kbps and fix the receiver 80 cm from the transmitter to measure the processing time and BER of both schemes. Specifically, we increase the number of transmitters of the same shape and size from one to four in turn. The experimental results are presented in Figure 11 and Figure 12 in the form of a box plot and bar chart, respectively. Intuitively, the number of LEDs affects the extraction time because the proposed scheme requires the stitching of the adjacent blocks. This phenomenon is also demonstrated by the median times of the proposed scheme increasing from 0.23 ms to 0.59 ms with the rise in the number of LEDs, as shown in Figure 11. However, the grid virtual division scheme still has lower median times than the baseline ranging from 1.02 ms to 1.51 ms under the same circumstances, which strongly demonstrates the effectiveness of the proposed scheme.

As shown in Figure 12, we then calculate the BERs obtained by the receiver when sequentially changing the number of transmitters from one to four. It is evident from the figure that the BER increases slightly as the number of transmitters increases. To be precise, the variation range of the BER at the baseline is 0.16% to 0.28%, while the variation range of the proposed scheme is 0.20% to 0.32%. Intuitively, if our proposed method is highly consistent with the regions extracted by the baseline scheme, the BER results will not fluctuate much. The statistical results are consistent with our expectations and remain within a reasonable threshold, which are sufficient to demonstrate the feasibility and effectiveness of the proposed scheme. It also shows that the increase in the number of transmitters has less of an impact on BER.

## 5. Conclusions

To cope with the issue of the performance bottleneck of RoI extraction during decoding, in this paper, we present an OCC system based on a fast grid virtual division scheme, which is purely built on COTS devices to improve OCC performance further. Essentially, the proposed innovation scheme establishes a virtual grid of size n×m on the received image and then randomly samples several pixels within different grids without traversing each pixel to locate the regions with high grayscale values quickly. As a result, the lightweight grid virtual division scheme effectively diminishes the decoding delay and strengthens the communication capacity of OCC. Furthermore, we have implemented the prototype of the proposed OCC system to evaluate its performance further. The experimental results clearly show that our proposed scheme can reduce the decoding delay by 70% compared to the baseline at a transmission frequency of 5 kHz, which powerfully demonstrates the effectiveness of the proposed innovation scheme. We believe that applying the simple yet delicate RoI extraction scheme to the OCC system can significantly reduce the decoding delay and largely broaden the application scenarios of OCC.

## Figures and Tables

**Figure 1 sensors-22-08375-f001:**
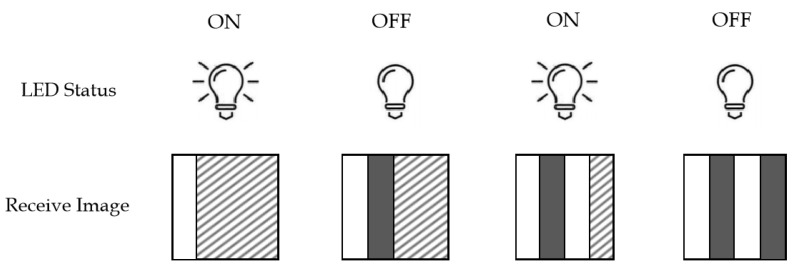
Schematic of the rolling shutter effect.

**Figure 2 sensors-22-08375-f002:**
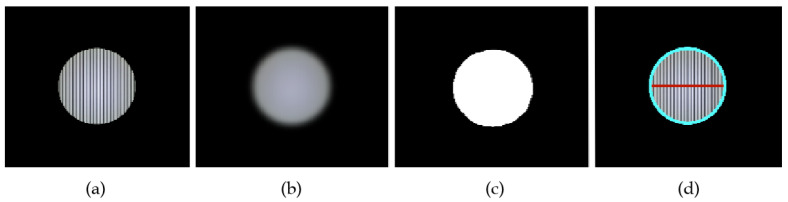
RoI extraction steps based on computer vision technology: (**a**) Original Image. (**b**) Gaussian Blur. (**c**) Image Binarization. (**d**) Contour Recognition and Pixel Sampling.

**Figure 3 sensors-22-08375-f003:**
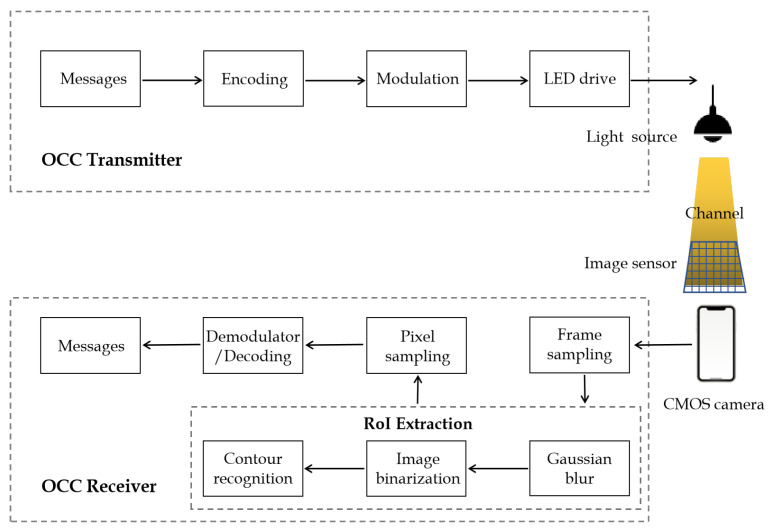
The schematic block diagram of an OCC system.

**Figure 4 sensors-22-08375-f004:**
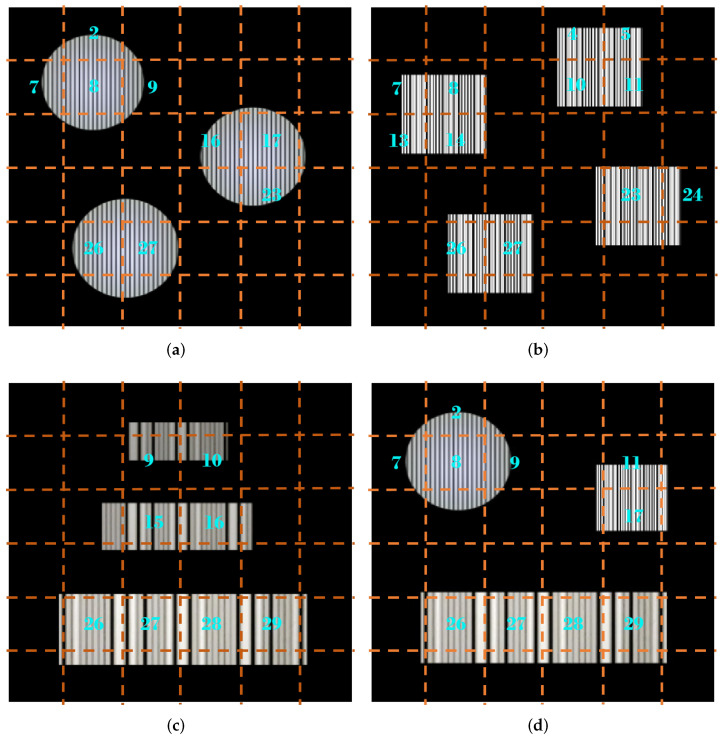
Region of interest projected by transmitters of different shapes in a frame: (**a**) Round LED luminaires. (**b**) Square LED luminaires. (**c**) Rectangular LED luminaires. (**d**) LED luminaires in a variety of shapes.

**Figure 5 sensors-22-08375-f005:**
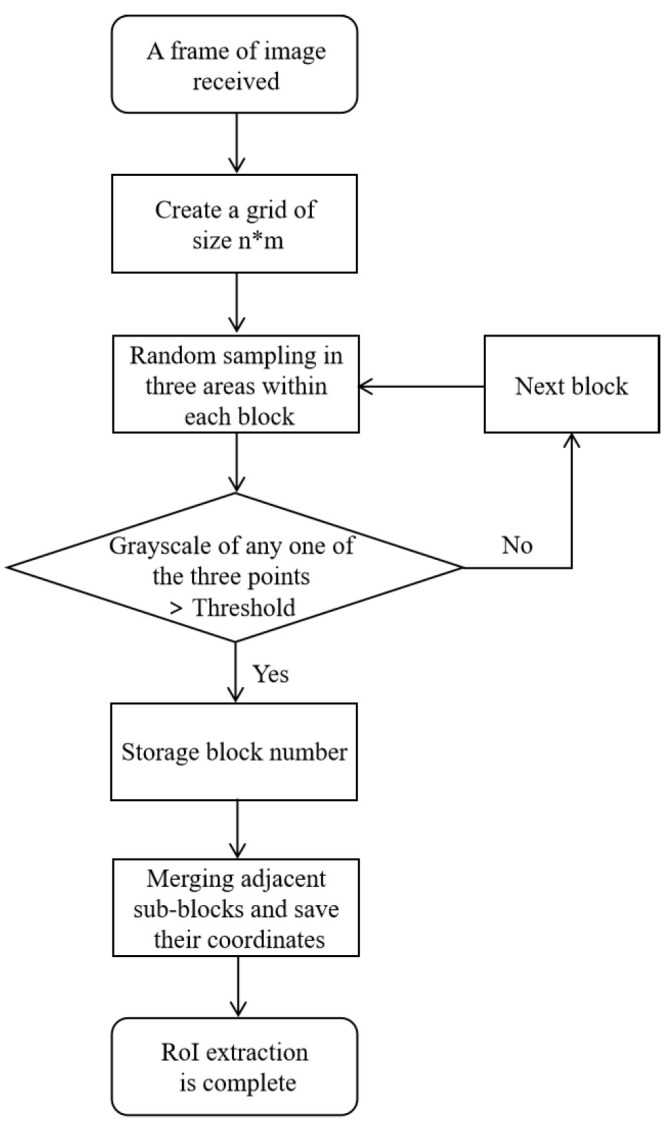
Flow chart of the proposed algorithm.

**Figure 6 sensors-22-08375-f006:**
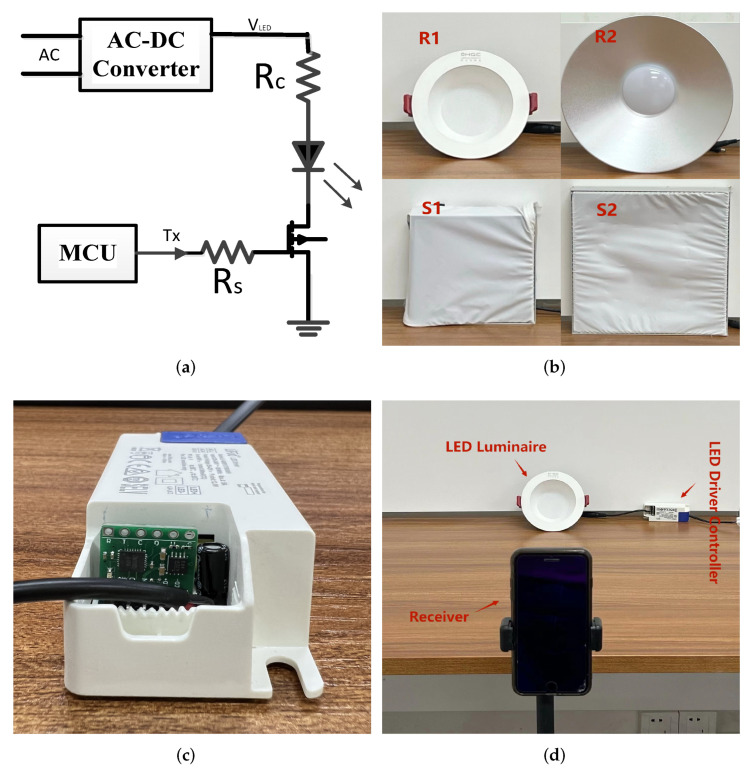
The experiment setup of the proposed OCC system. (**a**) Modulation circuit diagram for modulation circuitry. (**b**) LED transmitters in different shapes and sizes. (**c**) LED driver controller. (**d**) Experimental scenario for the proposed OCC system.

**Figure 7 sensors-22-08375-f007:**
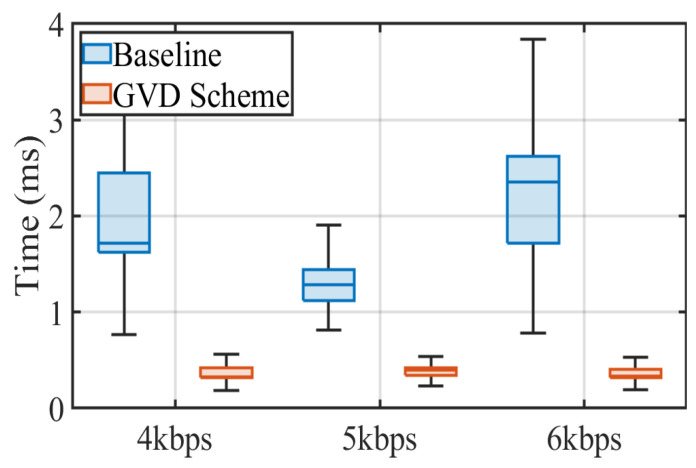
RoI extraction performance under variable transmission frequencies.

**Figure 8 sensors-22-08375-f008:**
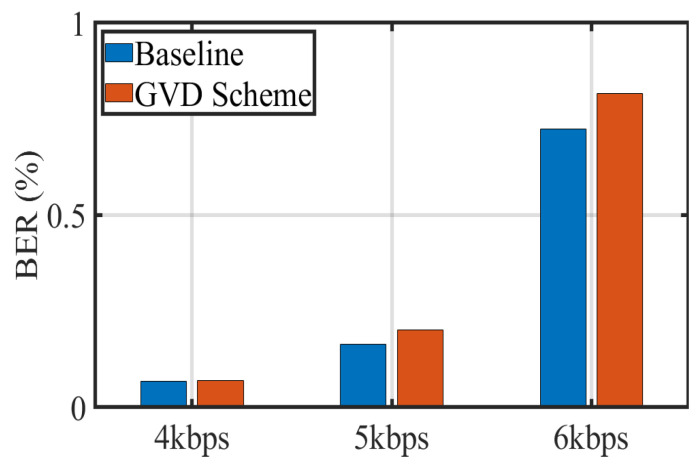
BER with variable transmission frequency.

**Figure 9 sensors-22-08375-f009:**
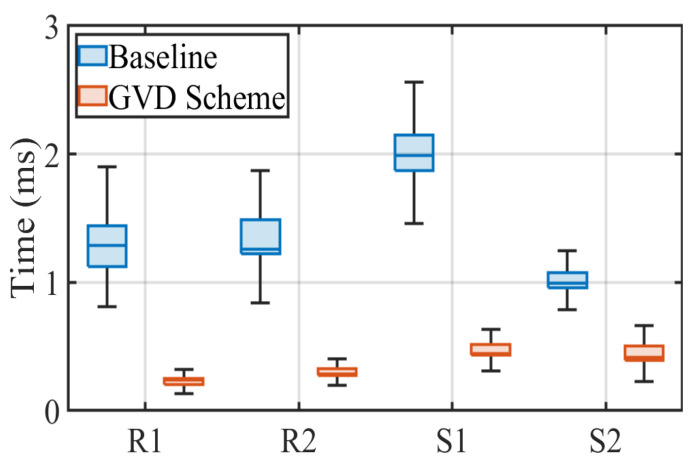
RoI extraction performance under variable LED shapes and sizes.

**Figure 10 sensors-22-08375-f010:**
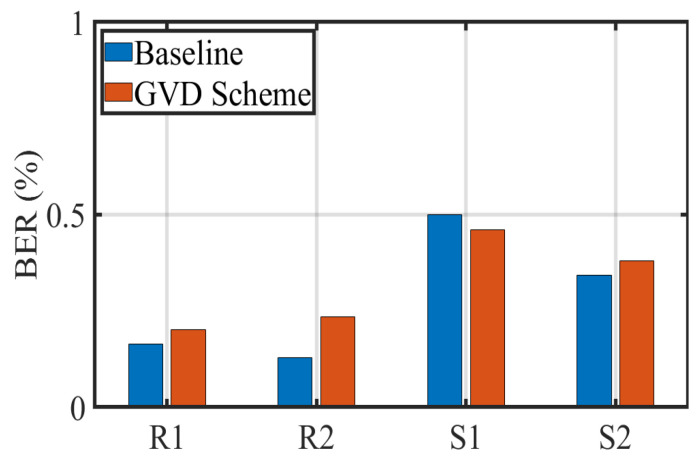
BER with variable LED shape and size.

**Figure 11 sensors-22-08375-f011:**
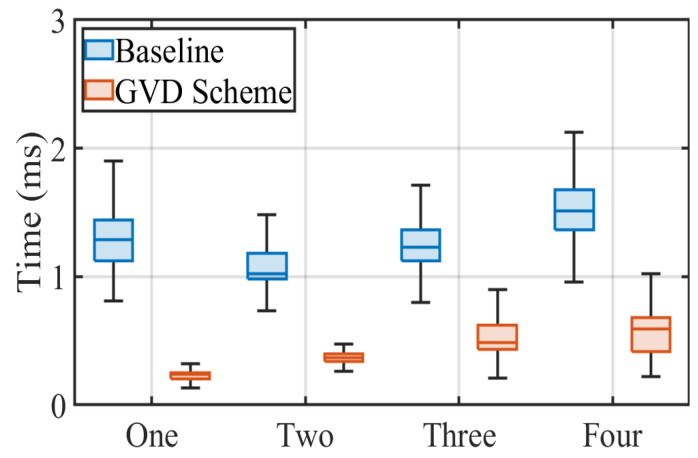
RoI extraction performance under variable LED number.

**Figure 12 sensors-22-08375-f012:**
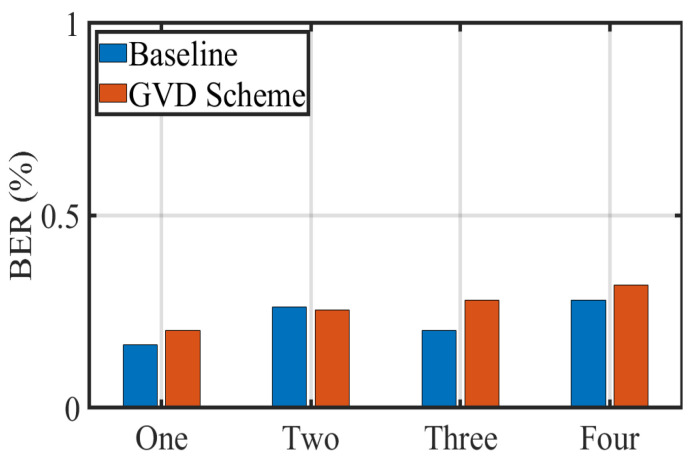
BER with variable LED number.

**Table 1 sensors-22-08375-t001:** Experiment parameters.

Parameter	Value
Round transmitter No. 1	0.13 m
Round transmitter No. 2	0.3 m
Square transmitter No. 1	0.14 m × 0.14 m
Square transmitter No. 2	0.25 m × 0.25 m
Microcontroller	ARM Cortex-M4 GD32F330G8U6
Modulation	On-Off Keying
Receiver	iPhone 8 Plus
Frame rate	30 fps
Exposure times	1/8000 s
ISO	350
Screen resolution	1920×1080
Pixel	dual 12-megapixel

## Data Availability

Not applicable.

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
