# Peer review of "High-Speed Extraction of Regions of Interest in Optical Camera Communication Enabled by Grid Virtual Division"

_sensors, 2022, doi:10.3390/s22218375_

Round 1

Reviewer 1 Report

This paper presents an algorithm for ROI detection in OCC communications based on grid division of the received image. The results show a speed improvement in comparison with conventional techniques.

I have some concerns and comments about the manuscript I would like to share with the authors:

The modulation scheme is not clear from the beginning (line 146). from lines 163 and 168, it seems the modulation is not ON OFF keying. Only in line 264 you mention the modulation. Please describe the modulation scheme and the modulation circuit. 

In line 174 you mention "…Since the light emitted by the transmitter is brighter, the grayscale value of the pixel in the RoI projected by the transmitter in the received image is higher than the surrounding pixels."

This occurs only in one's transmissions. In the zeros transmission, the value is zero. Unless you perform the blurring process, you need to consider this issue. Please describe more detailed this assumption.

There is an error in figure 4 (d) since you have transmissions bands in different directions (11,17,23,29 have horizontal bands, while the others have vertical ones). This is impossible in a Rolling Shutter camera since all the bands should be in the same direction.

In step 2 pixel sampling, you mention "After the virtual grid is established, the decoder randomly samples the grayscale values of several pixels within different blocks."

This process should be explained in more detail, presenting descriptions of what distribution is used for the sample selection, how many points, the dependence on the block sizes, and how the process assures the ROI detection in the block. In my opinion, this is a crucial point for the justification of the results.

The meaning of the sentence in line 215 "specifically, if there is a result of the operation of adding m" is not clear enough. Please detail the description of the operation performed.

From the image composition process can be seen that there are parts of the ROI not considered. For example, in figure 4 (d) only blocks 11,17,23 and 29 are considered, but there is also part of the ROI in blocks 5, 6, 12, 18, 24 and 30. Please provide a more in-depth analysis of the aspect and the effects on the system performance. 

I recommend changing the algorithm presentation using a flow chart representation instead of a programing language sequence.

Subsection 4.2.1. Impact of Varying Transmission Frequencies deals with the ROI extraction performance when the transmission frequency changes, but you do not explain why it affects the ROI algorithm. Intuitively frequency should not affect the ROI detection. Please provide a discussion on this issue. Indeed in figure 6 the ROI detection time does not change for the scheme you propose but, yes, for the baseline. How do you explain this?

Something similar should be done with the BER results. You should explain why the ROI detection process affects the BER. Probably due to the parts of the ROI the proposed scheme does not consider. Please comment on your system's tradeoff between ROI detection time and BER.

Please use the typical X*10^-z format instead of % for BER values.

Reviewer 2 Report

This manuscript describes  a fast grid virtual division scheme based on pixel grayscale values to reduce the decoding delay and improving the communication capacity of OCC. The experimental results show that the decoding delay of the proposed scheme is 70% lower than that provided by the Gaussian blur scheme for the iPhone receiver at a transmission frequency of 5 kHz. We have some suggestions to the authors as below.

(1)When multiple LEDs are captured in the received image, it means that 1. The sizes of these LEDs are small or 2. LEDs are far away from the CMOS sensor. These two conditions will undoubtedly reduce the communication rate. Whether the author should compare the performance of the single LED that has a large area?

(2)It is suggested that the introduction of the paper focus on ROI extraction technology, compare current ROI extraction technology as much as possible, and explain their advantages and disadvantages.

(3)In this article, only proposed the ROI extraction technology can not support the content of a journal paper.

(4)Directly adding (or averaging) the received frames can also determine the light source. You can try to compare with the algorithm in this paper.

Round 2

Reviewer 1 Report

Thank you for your response. I think the paper has improved considerably.

Author Response

Thank you for your comments on the article.

Reviewer 2 Report

11. In the revised version, the author mentioned some current target detection algorithms. In the experiment, the algorithm in this paper should be compared with YOLO, transformer and other algorithms. The objects compared in the experiment are a little insufficient.

22. The algorithm in this paper seems to lack complexity analysis.
